# Transcriptomic analysis reveals high *ITGB1* expression as a predictor for poor prognosis of pancreatic cancer

Yosuke Iwatate[1], Hajime Yokota[2], Isamu Hoshino[3]*, Fumitaka Ishige[1], Naoki Kuwayama[3], Makiko Itami[4], Yasukuni Mori[5], Satoshi Chiba[1], Hidehito Arimitsu[1], Hiroo Yanagibashi[1], Wataru Takayama[1], Takashi Uno[2], Jason Lin[6], Yuki Nakamura[6], Yasutoshi Tatsumi[6], Osamu Shimozato[6], Hiroki Nagase[6]

1 Division of Hepato-Biliary-Pancreatic Surgery, Chiba Cancer Center, Chiba, Japan, 2 Department of Diagnostic Radiology and Radiation Oncology, Graduate School of Medicine, Chiba University, Chiba, Japan, 3 Division of Gastroenterological Surgery, Chiba Cancer Center, Chiba, Japan, 4 Division of Clinical Pathology, Chiba Cancer Center, Chiba, Japan, 5 Graduate School of Engineering, Faculty of Engineering, Chiba University, Chiba, Japan, 6 Laboratory of Cancer Genetics, Chiba Cancer Center Research Institute, Chiba Cancer Center, Chiba, Japan

* ihoshino@chiba-cc.jp

**Data Availability Statement:** All RNA-seq results are available from Gene Expression Omnibus (accession number GSE196009).

## Abstract

Transcriptomic analysis of cancer samples helps identify the mechanism and molecular markers of cancer. However, transcriptomic analyses of pancreatic cancer from the Japanese population are lacking. Hence, in this study, we performed RNA sequencing of fresh and frozen pancreatic cancer tissues from 12 Japanese patients to identify genes critical for the clinical pathology of pancreatic cancer among the Japanese population. Additionally, we performed immunostaining of 107 pancreatic cancer samples to verify the results of RNA sequencing. Bioinformatics analysis of RNA sequencing data identified *ITGB1* (Integrin beta 1) as an important gene for pancreatic cancer metastasis, progression, and prognosis. *ITGB1* expression was verified using immunostaining. The results of RNA sequencing and immunostaining showed a significant correlation (r = 0.552, p = 0.118) in ITGB1 expression. Moreover, the ITGB1 high-expression group was associated with a significantly worse prognosis (p = 0.035) and recurrence rate (p = 0.028). We believe that ITGB1 may be used as a drug target for pancreatic cancer in the future.

## Introduction

Pancreatic cancer is a lethal cancer type with a poor prognosis and severe recurrence rate. It has the fourth and seventh highest cancer-related mortality rate in Japan and the world, respectively [1,2]. The overall five-year survival rate of pancreatic cancer is 10%, and it increases to only 20% even after curative surgery, making it one of the most lethal cancer types [3–5]. Unfortunately, there are no established sensitive markers for predicting the recurrence and survival of pancreatic cancer, and no therapeutic target gene has been determined. Technological development has facilitated the understanding of cancer genomics, and high-

**Funding:** The author(s) received no specific funding for this work.

**Competing interests:** NO authors have competing interests.

throughput gene expression analysis has revolutionized cancer genetics in the last 15 years. Even for pancreatic cancer, large-scale genome analyses with next-generation sequencing (NGS) have been performed [6]. Transcriptomic analyses on a large sample size classified RNA signatures of pancreatic cancer into classical and basal-like types, and further into four subtypes: squamous, pancreatic progenitor, immunogenic, and aberrantly differentiated endocrine exocrine [7,8]. In recent years, public databases such as The Cancer Genome Atlas Program (TCGA) and Gene Expression Omnibus have been constructed, and the gene expression data obtained from them are of great value for understanding the molecular mechanism, diversity, diagnosis, and clinical outcomes of cancers, including pancreatic cancer.

However, transcriptomic analysis of pancreatic cancer samples from the East Asian and Japanese population are lacking. To understand and analyze the mechanism and molecular markers of pancreatic cancer among the Japanese population, we performed a transcriptomic analysis in 12 Japanese patients with pancreatic cancer and compared the results with the TCGA data. The target genes thought to be involved in prognosis and recurrence of pancreatic cancer were narrowed down. We aimed through this study to clarify the relationship between the expression of the target gene by sequencing and the protein expression by immunostaining, where the expression of the target gene was further verified through immunostaining of a large number of patient samples. We identified *ITGB1* as an important gene in the progression of pancreatic cancer. Our findings suggest that a high *ITGB1* expression could predict the prognosis and recurrence of pancreatic cancer. ITGB1 is a constituent of β subunits in integrin molecules [9]. Integrin is mainly present in the plasma membrane and plays a role in cell-cell adhesion, cell-extracellular matrix adhesion, and signal transduction [9,10]. The lack of such adhesion leads to the withdrawal of cell survival signals, resulting in an exfoliation-induced apoptotic process called "anoikis" [11]. Cancer cells are resistant to anoikis through certain integrin types, which is one of the key mechanisms for successful infiltration, migration, and metastasis [11]. It has been reported that high *ITGB1* expression significantly correlated with the deterioration of prognosis in colorectal, breast, and lung cancers, but its correlation with pancreatic cancer remains controversial [12–17].

## Materials and methods

### Study population criteria

Between January 2013 and May 2018, 138 patients diagnosed with pancreatic ductal adenocarcinoma (PDAC) after its surgical removal without neoadjuvant chemotherapy and preoperative radiation were included in the study at the Chiba Cancer Center in Japan. Total RNA was extracted from 15 patients, including nine frozen specimens stored in our institute's biobank, and comprehensively analyzed by NGS. This study was approved by the Chiba Cancer Center Review Board (grant number H29-006). All procedures followed were in accordance with the ethical standards of the responsible committee on human experimentation and with the Helsinki Declaration of 1964 and its later amendments. Written informed consent was obtained from the patients for publication of this study and the accompanying clinicopathological data.

### RNA sequencing

Total RNA was isolated from a frozen tissue block containing approximately 50–100 mg of PDAC tissue using the miRNeasy Mini Kit (Qiagen) according to the manufacturer's instructions. Samples with an RNA integrity number (RIN value) of 7.0 or higher were used for RNA sequencing. The library for NGS was built with the Ion Proton ™ equipment (Thermo Fisher Scientific) using a 2 × 75 base pair (bp) pair-end protocol. Eight libraries were sequenced, and 34–60 million pairs were generated. The number of reads mapped to the annotated genomic

function was quantified from the BAM file using the function number of the Subread package (http://subread.sourceforge.net/). Differential expression was determined via linear modeling based on Bioconductor (ver3.11) and the linear model for microarray data (LIMMA) [18]. Genes with p values <0.0001 were considered as "differential expressed genes" (DEGs), and gene set enrichment analysis (GSEA) was performed (https://www.gsea-msigdb.org/gsea/index. jsp). Pathway analysis was performed using the Kyoto Encyclopedia of Genes and Genomes (KEGG). Additionally, we analyzed the protein–protein interactions of DEGs and visualized them with Cytoscape (ver 3.8.1) to identify the "hub genes." The hub genes were pre-evaluated using an online software named R2: Genomics Analysis and Visualization Platform (https://hgserver1.amc.nl/cgi-bin/r2/main.cgi) using the gene expression and prognostic data from TCGA. To assess whether the expression of the selected hub gene correlates with other clinicopathological factors, including prognosis, immunohistochemistry (IHC) was used for verification.

## Immunohistochemical analysis of ITGB1

ITGB1 levels were measured by IHC using mouse monoclonal anti-human ITGB1 protein antibody (4B7R, 1:100; Santa Cruz Biotechnology, Dallas, TX, USA). Five micrometers thick sections were obtained from formalin-fixed, paraffin-embedded tissues using a VEN-TANA Optiview DAB Universal Kit (Roche, Basel, Switzerland) and a VENTANA Bench-Mark ULTRA automated slide stainer (Roche, Bazel, Switzerland). Enzyme-induced antigen retrieval was performed using ISH Protease 1 (Roche, Basel, Switzerland) for 32 min at 36°C, and the primary antibody of ITGB1 was applied to the sample for 120 min at 36°C.

The percentage of stained tumor cells and the intensity of the staining for ITGB1 were evaluated by two pathologists. The expression status of these proteins (low/high) was determined by the IHC score as the product of the percentage and intensity of tumor cells with any membrane staining.

## IHC scoring of ITGB1 and related definitions

ITGB1 staining is generally observed in vascular smooth muscle tissue, and the levels of staining in this area were considered as controls. In addition, the percentage of tumor cells stained was scored as follows: 0%, 0; >0 to ≤20%, 1; >20% to ≤40%, 2; >40% to ≤60%, 3; >60% to ≤80%, 4; >80%, 5; and 100%, 6. The staining intensity of tumor cells was scored from 0 to 3 as follows: no staining at all, 0; staining at an intensity lower than the control, 1; staining at the same level as the control, 2; staining at an intensity higher than the control, 3. The product of the scores for the percentage of stained tumor cells and the staining intensity was calculated, and *ITGB1* expression in IHC of the tumor cell tissue was scored.

Spearman's correlation coefficient values were used to examine the correlation between IHC expression scores and RNA-sequencing expression. Cases with an IHC expression score higher than the mean RNA expression level of *ITGB1* using a regression line were defined as the high expression group of *ITGB1*.

## Definitions of variables for clinicopathological factors and statistical analysis

The significance of the correlation between the RNA expression level of *ITGB1* using RNA-seq and the IHC score of ITGB1 using immunohistochemistry was evaluated using the Spearman's rank correlation coefficient (r, ρ). Furthermore, the significance of the difference between *ITGB1* expression and some clinical and pathological variables was calculated using the $\chi^2$ test,

Fisher's exact test, or the Mann–Whitney U test. Overall survival (OS) was defined as the time from surgery to the final observation of survival. Disease-free survival (DFS) was defined as the time between surgery and the confirmation of recurrence. Survival curves were created using the Kaplan–Meier method, and the log-rank test was used to assess significant differences and determine key factors. A multivariate analysis was performed using the Cox regression model. Statistical significance was set at p <0.05.

## Results

### Patient backgrounds

Between January 2013 to March 2018, 138 patients were pathologically diagnosed with PDAC after surgical removal. Of these, 114 patients underwent surgery without preoperative chemotherapy or radiation therapy. In three cases, intraductal papillary mucinous carcinoma (IPMC) with an infiltrative component was diagnosed, and the infiltration site was too small; therefore, the residual sample could not be evaluated. We excluded three cases because distant metastasis was detected during the operation or because it was complicated by multi-organ cancer. One more patient, who was referred from another hospital, was excluded because of recurrence of residual pancreatic cancer. Thus, a retrospective study was conducted on 107 of the 138 patients. The biobank at our hospital included frozen specimens for nine patients. For five patients, the biobank had stocked specimens of only cancer tissue but had both cancer and normal tissue stocked for the other four patients. Specimens for six other cases were obtained during the operation, making the total specimens available 15, of which, 10 were pairs and only 5 were cancer tissues. We attempted to extract RNA from 10 pairs of cancerous and normal tissues and five cases of cancer tissue alone. Out of those 10-pair specimens, only eight pairs and two cancer tissues passed the quality check with a RIN value ≥ 7.0. All five cases with only the cancer tissues showed RIN values ≥7.0. One pair of biobank specimens was excluded because both were possible normal pancreatic tissue. One pair of specimens obtained during the operation was excluded because both were possible cancer tissue. One pair of that was excluded because later, the pathological result was found to be adenosquamous carcinoma.

A total of 17 samples from 12 patients, including five pairs of cancer and normal tissues and seven samples of only cancer tissue, were subjected to NGS. RNA-sequencing results were verified by IHC using the above 107 samples. The observation period was from January 2013 to July 2020, with a median period of 804 days (58–2,481 days). The median age was 70 years (50–87 years). The male-to-female ratio was 60:47 (Table 1). Curative resection R0 occurred in 89 cases, and the histological types were good, moderate, and poor in 46, 53, and 8 cases, respectively (Table 1). Lymph node metastasis was observed in 76 patients. Among the common T-factors in the TNM classification by The Union for International Cancer Control (UICC) (8th edition), T2 (2 cm< max tumor diameter ≤4cm) was the most common (60 cases). In the TNM classification (UICC 8th), stage III was the most common (39 cases), followed by stage II (37 cases) (Table 1).

### RNA sequencing

Among the 11,272 mapped mRNAs, 314 genes were differentially expressed in cancer tissues compared to the adjacent normal tissues (S1 Fig). When these genes were analyzed by the KEGG pathway analysis using GSEA, the significant pathways detected were (in order): Extracellular matrix (ECM)-receptor interaction, focal adhesion, protein digestion and absorption, phagosome, and the phosphatidylinositol 3-kinase-alpha serine/threonine-protein kinase (PI3K-Akt) signaling pathways (S1 Table). The top five pathways included 37 DEGs, including

**Table 1. Relationship between clinocopathological parameters and ITGB1 status.**

| | ITGB1 status | | | | | P value |
|---|---|---|---|---|---|---|
| **Expression type** | **low N (%)** | | **high N (%)** | | | |
| Sex | | | | | | |
| Male | 35 | (32.7%) | 25 | (23.4%) | | |
| Female | 32 | (29.9%) | 15 | (14.0%) | | 0.25* |
| Age | | | | | | |
| 70 (50–87) | 69 (51–83) | | 74 (50–87) | | | 0.037** |
| Preoperative CEA | | | | | | |
| 3.3 (0.5–47.3) | 3.1 (0.5–28.5) | | 3.4 (0.8–47.3) | | | 0.384** |
| Preoperative CA19-9 | | | | | | |
| 137.4 (0–47588.2) | 90.1 (0–19447) | | 481.8 (0–47588.2) | | | 0.039** |
| Operation type | | | | | | |
| PD | 46 | (43.0%) | 24 | (22.4%) | | |
| DP | 21 | (19.6%) | 14 | (13.1%) | | |
| TP | 0 | (0.0%) | 2 | (1.9%) | | 0.209*** |
| Cytology | | | | | | |
| negative | 59 | (55.1%) | 34 | (31.8%) | | |
| positive | 8 | (7.5%) | 6 | (5.6%) | | 0.65* |
| Margin status | | | | | | |
| R0 | 58 | (54.2%) | 31 | (29.0%) | | |
| R1 | 8 | (7.5%) | 8 | (7.5%) | | |
| R2 | 1 | (0.9%) | 1 | (0.9%) | | 0.423*** |
| Differenciation | | | | | | |
| well | 30 | (28.0%) | 16 | (15.0%) | | |
| moderate | 31 | (29.0%) | 22 | (20.6%) | | |
| poor | 6 | (5.6%) | 2 | (1.9%) | | 0.594*** |
| Interstitium type | | | | | | |
| int | 62 | (57.9%) | 36 | (33.6%) | | |
| med | 1 | (0.9%) | 0 | (0.0%) | | |
| sci | 4 | (3.7%) | 4 | (3.7%) | | 0.670*** |
| lympathic invasion | | | | | | |
| negative | 19 | (17.8%) | 10 | (9.3%) | | |
| positive | 48 | (44.9%) | 30 | (28.0%) | | 0.705* |
| vascular invasion | | | | | | |
| negative | 1 | (0.9%) | 0 | (0.0%) | | |
| positive | 66 | (61.7%) | 40 | (37.4%) | | 0.438* |
| neural invasion | | | | | | |
| negative | 4 | (3.7%) | 2 | (1.9%) | | |
| positive | 63 | (58.9%) | 38 | (35.5%) | | 0.722* |
| Lymph node metastasis | | | | | | |
| negative | 19 | (17.8%) | 12 | (11.2%) | | |
| positive | 48 | (44.9%) | 28 | (26.2%) | | 0.856* |
| p Max diameter (cm) | | | | | | |
| 3.3 (1.0–11.6) | 3.3 (1.5–8.5) | | 3.3 (1.0–11.6) | | | 0.821** |
| Postoperative adjuvant chemotherapy | | | | | | |
| yes | 15 | (14.0%) | 12 | (11.2%) | | |
| no | 52 | (48.6%) | 28 | (26.2%) | | 0.381* |
| pT (UICC) 8th | | | | | | |

*(Continued)*

**Table 1.** (Continued)

| Expression type | ITGB1 status | | | | P value |
|---|---|---|---|---|---|
| | low N (%) | | high N (%) | | |
| T1 | 12 | (11.2%) | 7 | (6.5%) | |
| T2 | 38 | (35.5%) | 22 | (20.6%) | |
| T3 | 17 | (15.9%) | 11 | (10.3%) | 0.971*** |
| pStage (UICC 8th) | | | | | |
| A | 7 | (6.5%) | 5 | (4.7%) | |
| B | 9 | (8.4%) | 5 | (4.7%) | |
| A | 3 | (2.8%) | 2 | (1.9%) | |
| B | 23 | (21.5%) | 14 | (13.1%) | |
| III | 25 | (23.4%) | 14 | (13.1%) | 0.996*** |

*The significance of the difference between ITGB1 and ITGAV expression and several clinical and pathologic variables was assessed by the $\chi^2$ test.

**The significance of the difference between ITGB1 and ITGAV expression and several clinical and pathologic variables was assessed by the Mann–Whitney U test.

***The significance of the difference between ITGB1 and ITGAV expression and several clinical and pathologic variables was assessed by Fisher's exact test.

ITGB1, *collagen 4 alpha 1 (COL4A1)*, *COL4A2*, *integrin alpha 5 (ITGA5)*, *integrin alpha V (ITGAV)*, *COL1A1*, and *COL1A2* (Fig 1). Network analysis using Cytoscape (ver. 3.8.0) was performed on these DEGs, and the hub gene was found to be *ITGB1* (Fig 1). Examination of the relationship between gene expression and prognosis using the R2 platform showed that *ITGB1* was significantly correlated with the prognosis of pancreatic cancer (p = 0.036), but *COL4A1* (p = 0.084), *COL4A2* (p = 0.121), and *ITGA5* (p = 0.285) were not (Fig 2). *ITGB1* expression (with p <0.05) was verified by immunostaining.

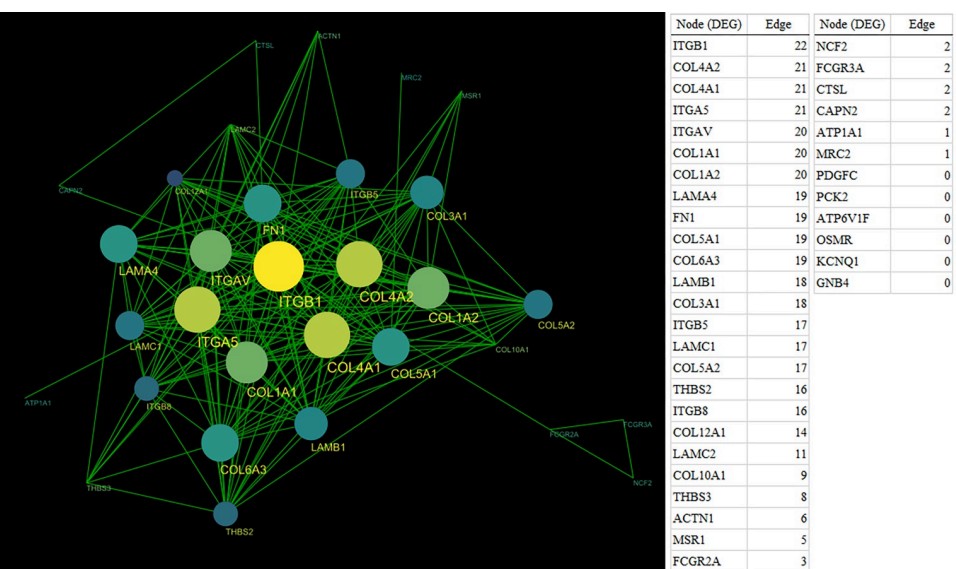

**Fig 1. Network analysis in the top five pathways.** In the DEGs mapped to the top five pathways, protein-protein interaction analysis was performed, and the network was constructed by Cytoscape (ver. 3.8.0). In this network, the DEGs are called nodes, and the correlated nodes are connected by lines called edges. Furthermore, in this network, the node with the most edges was called the hub gene, suggesting a clinically significant possibility. Second to the top hub genes with 22 and 21 edges, respectively, were selected from the network analysis. *ITGB1, COL4A1, COL4A2,* and *ITGA5* were detected.

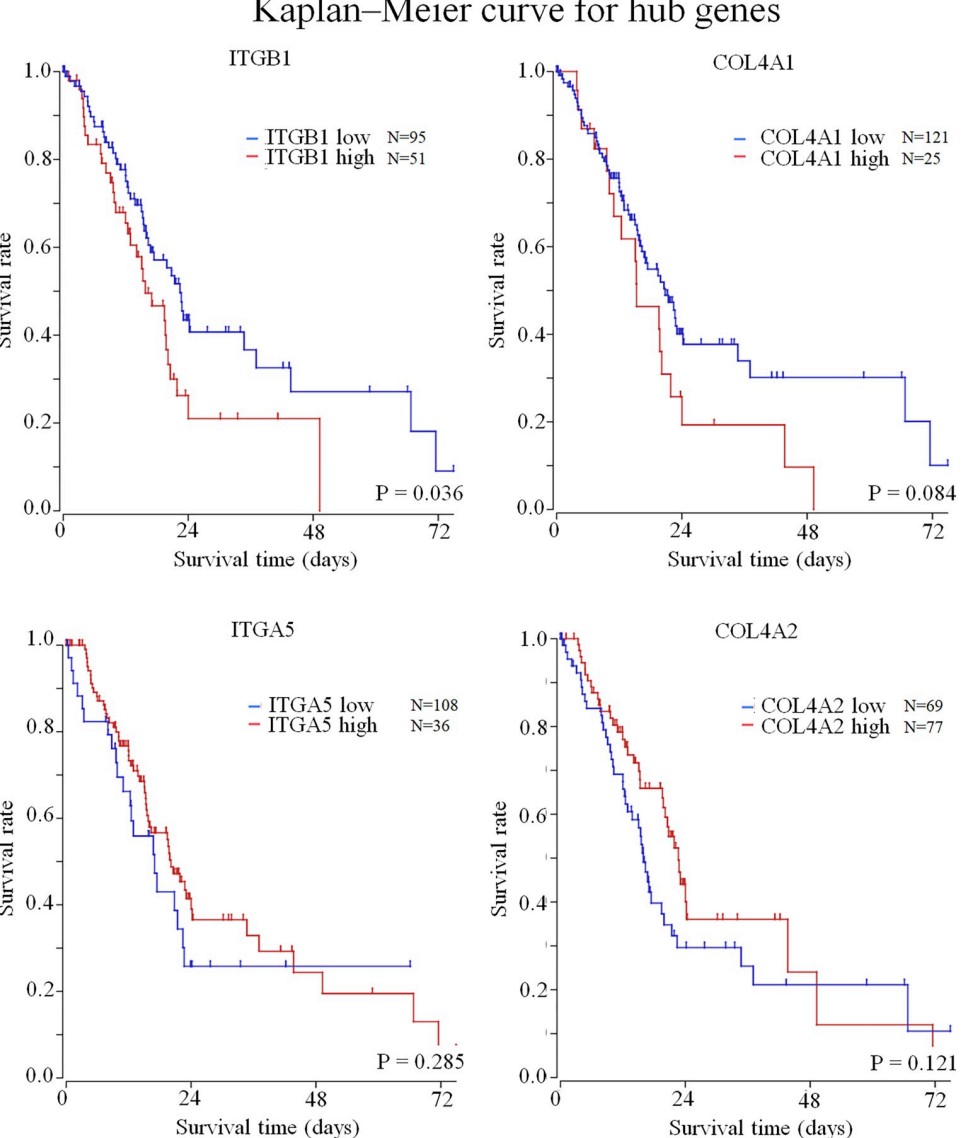

**Fig 2. Kaplan–Meier curve for pre-validation of the hub genes by the R2 platform.** The four hub genes detected by protein-protein interaction analysis were pre-verified for prognosis using the R2 platform, an open-source external databank. Prognostic analysis with the R2 platform using TCGA showed *ITGB1* to be significantly involved in the deterioration of prognosis (p = 0.036). The expression of *COL4A1*, *COL4A2*, and *ITGA5* was not significantly correlated with poor prognosis (p = 0.084, 0.121. 0285).

## IHC scoring of ITGB1

The stromal tissue of the tumor samples was stained uniformly for ITGB1 in all cases, with a slightly weaker intensity than that of the surrounding normal pancreatic tissue. The tumor cell IHC scores of ITGB1 were between 0–18 (median = 7) (Fig 3).

## Correlation between IHC score and RNA-sequencing

For ITGB1, the IHC score tended to correlate with RNA-seq expression, but the difference was not significant (r = 0.552, ρ = 0.476, p = 0.118). The median *ITGB1* expression level was 9.22. Since the IHC score corresponding to the median *ITGB1* expression level in NGS was 10.5 with the regression line, an IHC score ≥ 11 indicated high *ITGB1* expression (Fig 4).

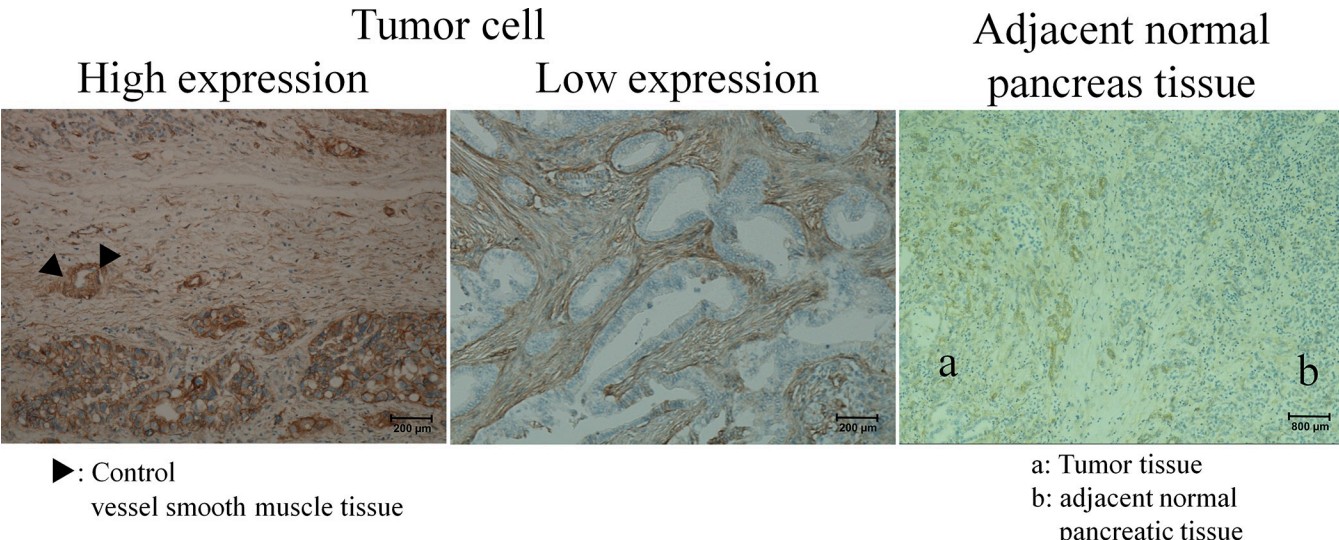

**Tumor cell**

**High expression**    **Low expression**    **Adjacent normal pancreas tissue**

▶: Control
 vessel smooth muscle tissue

a: Tumor tissue
b: adjacent normal
 pancreatic tissue

**Fig 3. IHC of ITGB1 in pancreatic cancer tissue.** For ITGB1 IHC analysis, the cell membrane and all vascular smooth muscle were stained in the positive control tissue. Staining levels in vascular smooth muscle were used as controls (Magnification = 160 X, 160 X, and 40 X, respectively).

### Relationship between IHC status and clinicopathological factors

High ITGB1 expression was observed in 40 patients (37.4%). The patients in the ITGB1 high-expression group were significantly older and had higher CA19-9 levels (p = 0.037 and 0.039, respectively), but other clinicopathological factors such as preoperative tumor marker levels and lymph node metastasis were not significantly different (Table 1).

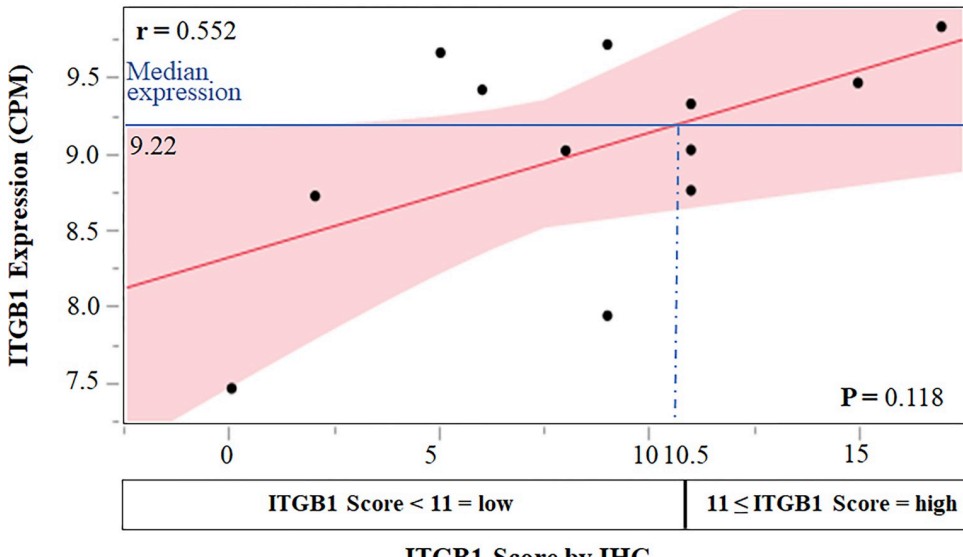

**Correlation between ITGB1 IHC scoring and RNA-seq**

r = 0.552

Median expression

9.22

P = 0.118

ITGB1 Score < 11 = low 11 ≤ ITGB1 Score = high

**ITGB1 Score by IHC**

**Fig 4. Correlation between IHGB1 IHC scoring and RNA-seq.** IHCs scores were set on the X-axis, the RNA-sequencing expression levels were set on the Y-axis, and the correlation was graphed. Although the IHC score and RNA-sequencing expression tended to have a correlation, it was not significant (r = 0.552, ρ = 0.476, p = 0.118). For these relationships, a regression line was created, and the IHC score corresponding to the median RNA-expression level was calculated to be 10.5. Therefore, an IHC score ≥11 signified high *ITGB1* expression.

### Relationship between clinicopathological factors and the prognosis and recurrence of pancreatic cancer

The presence of the tumor marker CA19-9 was associated with significantly worse OS and DFS (CA19-9: p = 0.003 and <0.001, respectively). Similarly, positive nerve infiltration, tumor diameter, T factor, and lymph node metastasis worsened both OS and DFS. In addition, surgical procedure, operation time, bleeding volume, vascular infiltration, and postoperative adjuvant chemotherapy group significantly correlated with OS, and the histological type and lymphatic vessel infiltration correlated with DFS. (Table 2). In the ITGB1 high-expression groups, the prognosis of pancreatic cancer, along with lymph node metastasis, T factor, and tumor markers, was significantly worse (p = 0.035). Likewise, the ITGB1 high-expression group showed a significantly worse recurrence rate (p = 0.028) (Table 2) (Fig 5).

### Evaluation of prognosis and recurrence predictors by multivariate analysis

All the following factors—ITGB1 expression, CA19-9, operation time, operation type, bleeding volume, vascular invasion, neural invasion, lymph node metastasis, tumor diameter, and T factor— were significantly associated with poor prognosis of pancreatic cancer. However, the tumor diameter was excluded because it was confounded with the T factor. Multivariate analysis performed with eight of these factors showed that ITGB1 expression, surgical procedure, nerve infiltration, T factor, and lymph node metastasis were independent prognostic factors (Table 3). Similarly, ITGB1, ITGAV, CA19-9, differentiation, lymphatic invasion, neural invasion, tumor diameter, T factor, and lymph node metastasis were all significantly correlated with the pancreatic cancer recurrence rate, and multivariate analysis with eight of these factors showed that ITGB1, neural invasion, T factor, and lymph node metastasis were independent recurrence factors (Table 3).

## Discussion

In this study, we were able to understand how the dynamics of gene expression in cancer tissues are associated with clinicopathological factors. We evaluated the expression of *ITGB1*, a factor that has been reported to contribute to the infiltration and metastasis of various carcinomas. Using transcriptome analysis of pancreatic cancer tissues, we confirmed *ITGB1* to be an independent prognostic factor in pancreatic cancer.

ITGB1 is a constituent of integrin molecules. It forms heterodimers with β subunits consisting of integrin β chains and α subunits consisting of integrin α chains [9]. Integrin is mainly present in the plasma membrane. It is involved in cell-cell adhesion, cell-extracellular matrix adhesion, and signal transduction. It has been confirmed that there are eight types of β subunits and 18 types of α subunits, and ITGB1 forms β subunits and dimers with various α subunits which adhere to collagen, fibronectin, and vitronectin. These connective tissue proteins, in turn, constitute the interstitium and laminin and form the basement membrane [9,10]. While it was reported that it functioned as a cell by "construction of scaffolds" with integrin and by "receive of survival signal" through adhesions with integrin, loss of these scaffolds causes an exfoliation-induced apoptotic process called "anoikis" [11]. Cancer cells have been reported to avoid "anoikis" through integrins, which are involved in proliferation, migration, infiltration, and metastasis [11]. In pancreatic cancer, some reports indicated that ITGB1 is distributed as α2β1 and α5β1 in tumor cells and binds to the basement membrane and extracellular matrix [19]. It also regulates cytokine secretion, activates intracellular signal transduction, causes cell proliferation and infiltration, and regulates protein production in the matrix [19,20].

High expression of ITGB1 is associated with a poor prognosis of colorectal, lung, and breast cancer, cancer recurrence, and cancer angiogenesis [12–17]. The same is true for pancreatic

**Table 2. Univariate analysis of prognostic factors with ITGB1 for OS and DFS.**

| Variable | No. of Patients (%) | Univariate analysis for OS | | Univariate analysis for DFS | |
|---|---|---|---|---|---|
| | | Median (95% confidence interval) | Log-Rank | Median (95% confidence interval) | Log-Rank |
| | | (days) | (P value) | (days) | (P value) |
| Gender | | | | | |
| Male | 60 (56.1) | 864 (622–1065) | 0.527 | 391 (260–575) | 0.760 |
| Female | 47 (43.9) | 990 (494–1324) | | 356 (223–498) | |
| Age | | | | | |
| ≥ 70 | 55 (51.4) | 1155 (730–1276) | 0.299 | 408 (282–561) | 0.949 |
| < 70 | 52 (48.6) | 804 (515–963) | | 277 (247–458) | |
| Follow-up (days) | | | | | |
| Median | 804 | | | | |
| Range | 58–2481 | | | | |
| preoperative CEA | | | | | |
| ≤ 3.3 | 54 (50.0) | 1156 (730–1324) | | 402 (279–737) | |
| > 3.3 | 53 (50.0) | 804 (572–911) | 0.110 | 302 (225–458) | 0.400 |
| preoperative CA-19-9 | | | | | |
| ≤ 137.4 | 54 (50.0) | 1175 (817–1487) | | 594 (373–777) | |
| > 137.4 | 53 (50.0) | 572 (393–866) | 0.003 | 252 (164–306) | < 0.001 |
| Operation type | | | | | |
| PD | 70 (65.4) | 730 (534–942) | | 307 (256–455) | |
| DP/TP | 37 (34.6) | 1324 (864–NA) | 0.007 | 458 (263–832) | 0.113 |
| Operation time | | | | | |
| ≤ 311 | 55 (51.4) | 1175 (817–1512) | | 428 (298–641) | |
| > 311 | 52 (48.6) | 711 (515–911) | 0.003 | 280 (243–498) | 0.087 |
| Bleeding volume | | | | | |
| ≥ 600 | 54 (50.5) | 1065 (7461512) | | 407 (282–575) | |
| > 600 | 53 (49.5) | 777 (560–990) | 0.043 | 280 (243–498) | 0.123 |
| Cytology | | | | | |
| CY0 | 91 (88.3) | 905 (746–1175) | | 395 (298–526) | |
| CY1 | 12 (11.7) | 454 (251–NA) | 0.159 | 208 (106–484) | 0.056 |
| Margin status | | | | | |
| R0 | 86 (83.5) | 864 (711–1175) | | 356 (268–498) | |
| R1/R2 | 17 (16.5) | 866 (455–1212) | 0.612 | 380 (135–575) | 0.117 |
| Differentiation | | | | | |
| Well | 47 (43.9) | 1187 (800–1512) | | 498 (282–839) | |
| Moderate/Poor | 60 (56.1) | 777 (534–963) | 0.055 | 298 (243–408) | 0.035 |
| Lymphatic invasion | | | | | |
| Negative | 29 (28.0) | 1243 (746–1881) | | 746 (282–NA) | |
| Positive | 78 (72.0) | 817 (615–979) | 0.122 | 304 (253–408) | 0.001 |
| Neural invasion | | | | | |
| Negative | 6 (6.5) | NA (1175–NA) | | NA (280–NA) | 0.020 |
| Positive | 101 (93.5) | 817 (656–990) | 0.022 | 356 (263–458) | |
| Vascular invasion | | | | | |
| Negative (v0/1) | 21 (19.6) | 1512 (560–NA) | | 455 (279–1064) | |
| Positive (v2/3) | 85 (80.4) | 823 (711–990) | 0.039 | 356 (256–484) | 0.147 |
| Interstitium type | | | | | |
| int | 98 (91.6) | 905 (735–1156) | | 360 (279–484) | |

*(Continued)*

**Table 2.** (Continued)

| Variable | No. of Patients (%) | Univariate analysis for OS | | Univariate analysis for DFS | |
|---|---|---|---|---|---|
| | | Median (95% confidence interval) | Log-Rank | Median (95% confidence interval) | Log-Rank |
| | | (days) | (P value) | (days) | (P value) |
| med + sci | 9 (8.4) | 396 (248–1881) | 0.223 | 209 (57–NA) | 0.807 |
| p Max diameter (cm) | | | | | |
| ≤ 3.3 | 57 (50.0) | 1155 (804–1324) | | 561 (312–777) | |
| >3.3 | 50 (50.0) | 711 (454–866) | 0.033 | 263 (160–356) | 0.003 |
| Lymph nodes | | | | | |
| Negative | 31 (29.0) | 1512 (990–NA) | | 962 (455–NA) | |
| Positive | 76 (71.0) | 735 (534–866) | < 0.001 | 271 (209–373) | < 0.001 |
| T factor (UICC 8th) | | | | | |
| T1/2 | 79 (76.7) | 990 (804–1276) | | 455 (302–641) | |
| T3 | 28 (26.2) | 541 (304–864) | 0.010 | 217 (144–343) | 0.024 |
| Postoperative adjuvant chemotherapy | | | | | |
| yes | 80 (74.8) | 942 (804–1243) | | 395 (298–526) | |
| no | 27 (25.2) | 599 (248–963) | 0.045 | 225 (106–455) | 0.143 |
| Stage (UICC 8th) | | | | | |
| A | 12 (11.2) | | | | |
| B | 14 (13.1) | | | | |
| A | 5 (4.7) | | | | |
| B | 37 (34.6) | 1187 (817–1487 (I,II)) | < 0.001 | 498 (312–764 (I,II)) | < 0.001 |
| III | 39 (36.4) | 560 (362–823 (III)) | (I,IIvsIII) | 256 (160–386 (III)) | (I,IIvsIII) |
| ITGB1 status | | | | | |
| low | 67 (62.6) | 1155 (800–1276) | | 458 (308–737) | |
| high | 40 (37.4) | 656 (447–866) | 0.035 | 247 (170–312) | 0.028 |

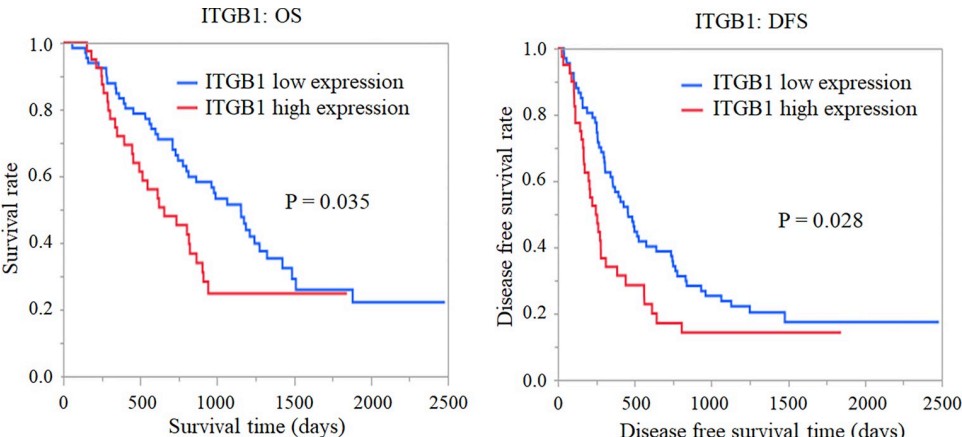

**Fig 5. Kaplan–Meier curve for overall survival and recurrence-free survival in ITGB1.** We performed immunostaining for ITGB1 in 107 patients diagnosed with PDAC who underwent radical resection without preoperative chemotherapy. Forty cases had an IHC score ≥11. The Kaplan–Meier curves for overall and recurrence-free survival are shown. IHC scores ≥11 significantly worsened survival time and recurrence-free survival time (p = 0.035, 0.028).

**Table 3. Multivariate analysis of prognostic factors for OS and DFS.**

| Variables | OS | | | DFS | | |
|---|---|---|---|---|---|---|
| | Hazard Ratio | 95% Confidence Limit | P value | Hazard Ratio | 95% Confidence Limit | P value |
| preoperative CA-19-9 | | | | | | |
| ≤ 137.4 (n = 54) | 1 | | | 1 | | |
| > 137.4 (n = 53) | 1.761 | 1.065–2.932 | 0.028 | 1.717 | 0.981–3.020 | 0.058 |
| Operation type | | | | | | |
| PD (n = 70) | 1 | | | | | |
| DP/TP (n = 37) | 0.393 | 0.199–0.768 | 0.006 | | NA | |
| Operation time | | | | | | |
| ≤ 311 (n = 55) | 1 | | | | | |
| > 311 (n = 52) | 1.027 | 0.547–1.883 | 0.930 | | NA | |
| Differentiation | | | | | | |
| Well (n = 47) | | | | 1 | | |
| Moderate/Poor (n = 60) | | NA | | 1.273 | 0.770–2.409 | 0.303 |
| Lymphatic invasion | | | | | | |
| Negative (n = 29) | | | | 1 | | |
| Positive (n = 78) | | NA | | 1.805 | 0.875–3.999 | 0.112 |
| Neural invasion | | | | | | |
| Negative (n = 6) | 1 | | | 1 | | |
| Positive (n = 101) | 3.960 | 1.086–25.789 | 0.035 | 5.323 | 1.358–36.153 | 0.014 |
| Lymph node | | | | | | |
| negative (n = 31) | 1 | | | 1 | | |
| positive (n = 76) | 2.694 | 1.464–5.254 | 0.001 | 3.015 | 1.560–4.760 | <0.001 |
| T factor (UICC 8th) | | | | | | |
| T1/2 (n = 79) | 1 | | | 1 | | |
| T3 (n = 28) | 2.326 | 1.317–4.014 | 0.004 | 2.126 | 1.171–3.794 | 0.014 |
| ITGB1 status | | | | | | |
| low (n = 64) | 1 | | | 1 | | |
| high (n = 43) | 1.912 | 1.102–3.297 | 0.022 | 1.914 | 1.110–3.262 | 0.020 |

cancer, and a few studies reported that high protein and gene expression of ITGB1 is positively correlated with a poor cancer prognosis [21–26]. A meta-analysis was performed by summarizing these studies [27]. The meta-analysis summarized reports of the association between ITGB1 expression and prognosis. Among the accumulated reports, two reports of immunohistochemical staining for pancreatic cancer were found [25,26]. Of these, the study by Sawai et al. used 78 pancreatic cancer patient specimens and investigated the association between ITGB1 expression and prognosis by immunohistological staining [26]. Their results, unlike ours, did not show a significant correlation between ITGB1 expression and prognosis [26]. However, in their report, about 20 postoperative cases of stage IV simultaneous liver metastasis were included, and the background of the patients was significantly different from that of ours, which targeted radical resection cases [26]. Also, the method of evaluating immunohistological staining was different between us and them. Yang et al. Targeted only R0, R1 resectable pancreatic cancer, and our study was consistent with the target cases [25]. In addition, as a result of investigating the relationship between ITGB1 expression and prognosis in 63 cases, the prognosis was poor in the high expression group as in our result [25]. In our study, the number of target cases was about twice as many, and the results conformed to their results.

The known preoperative tumor markers CA19-9 and CEA, which are potential prognostic factors, have cutoffs of 37 U/mL and 3 U/mL or 5 U/mL, respectively [28]. It was reported that high preoperative marker levels can be utilized as prognostic factors but not as therapeutic targets. In this study, we divided the median into two groups, and the inspection cutoffs for CA19-9 and CEA at our facility were 37 U/mL and 5 U/mL, respectively. When examined, high CA19-9 values were significantly correlated with DFS (OS; CA19-9>37.0, CEA>3.0, CEA>5.0 p = 0.051, 0.079, and 0.233, respectively, DFS; CA19-9>37.0, CEA>3.0, CEA>5.0 p = 0.001, 0.286, and 0.356, respectively), but in the multivariate analysis, CA19-9 was not an independent factor (p = 0.141). The high expression of the ITGB1 protein in tumor cells is reported to be 32.4% [21], but no clear threshold is known for the high and low expression of ITGB1 in cancer tissues [29–33].

In this study, deterioration of OS and DFS was observed in the high ITGB1 expression group of tumor cells. In the interstitial area, staining was uniformly observed in all cases, and the pancreatic stellate cells and fibroblasts were stained. In pancreatic cancer, it has been reported that high expression of ITGB1 is involved in the migration of cancer cells [34], and that high expression of ITGB1 and ITGA3 confers resistance to gemcitabine by increasing integrin α3β1 signaling [35]. Furthermore, integrin is involved in promoting infiltration and metastasis in lung and breast cancer; hence, the therapeutic strategies targeting integrins for cancer treatment are being developed [36,37]. Integrin-targeted treatment may facilitate the treatment of pancreatic cancer, as ITGB1 has been reported to inhibit cell proliferation, infiltration, and migration in pancreatic cancer [24,37–40].

Various integrin antagonists, such as α5β1 and αVβ3, are in the research stages, and their antitumor effects have been reported in breast cancer in vitro [29,41]. In clinical trials, there are currently no reports showing that a single integrin inhibitor is effective, but they are expected to be effective in combination with multiple agents such as immune checkpoint inhibitors (NCT00195278 and NCT04508179) [42,43]. It is expected that ITGB1 will contribute to markers and treatment in pancreatic cancer if the development, research, and clinical application of these drugs progress in the future.

However, our results were different from another similar study. The transcriptome study by Bailey *et al*. focused on specimens with ≥40% tumor cells and performed deep sequencing of 40% or less, and clarified the relationship between gene mutation and gene expression, and clustered and typed gene expression patterns, and *KRAS*, *TP53*, *CDKN2A*, and *SMAD4* were defined as gene mutations by exome analysis [6,8]. Their pathway analysis in exsome results are also different from those of our study, showing the WNT signaling, TGF-β signaling, and cell cycle as the top pathways [8]. A meta-analysis of PDAC transcriptome analysis including tissue microarray demonstrated results that are similar to ours, reporting that ECM-receptor interaction, PI3K-Akt signaling pathway, focal adhesion, and cancer pathways were significant pathways as well [44,45]. We assume that this difference in pathway analysis might be attributed to the fact that our analysis was influenced by the interstitium as the tumor tissue has more stroma compared to the normal tissues surrounding the tumor. Furthermore, our study had some limitations. The small number of samples used for sequencing may have been insufficient to verify the correlation with IHC. Furthermore, this was a retrospective study conducted at a single institution. For more accurate results, future studies need to be conducted prospectively and with more samples.

## Conclusion

Bioinformatics analysis of RNA sequencing data for pancreatic cancer identified *ITGB1* as an important hub gene. Immunohistochemical staining with multiple samples showed that both

DFS and OS were significantly shorter in the groups showing high *ITGB1* expression and were independent predictors of prognosis and recurrence in multivariate analysis. In addition, there has never been a report showing a causal relationship between mRNA expression by NGS and protein expression by IHC in the gene expression of ITGB1 for PDAC, and this may also be a significant report.

## Supporting information

**S1 Fig. Heatmap for 314 DEGs.** Genes with differential expression between the PDAC tissue and its adjacent pancreatic tissue were mapped and visualized on a heat map.
(TIF)

**S1 Table. Results for KEGG pathway analysis.** KEGG pathway analysis was performed on 314 DEGs, and a P value of 0.01 or less was considered significant, and 20 pathways were detected.
(XLSX)

## Author Contributions

**Conceptualization:** Yosuke Iwatate, Isamu Hoshino.

**Data curation:** Yosuke Iwatate.

**Formal analysis:** Yosuke Iwatate, Hajime Yokota, Makiko Itami.

**Investigation:** Yosuke Iwatate, Fumitaka Ishige, Naoki Kuwayama, Yasukuni Mori, Satoshi Chiba, Hidehito Arimitsu, Hiroo Yanagibashi, Yuki Nakamura, Yasutoshi Tatsumi, Osamu Shimozato.

**Methodology:** Fumitaka Ishige, Naoki Kuwayama, Wataru Takayama, Jason Lin.

**Supervision:** Takashi Uno, Hiroki Nagase.

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
