## [Decision Letter · Decision Letter 0]

4 Jan 2022

PONE-D-21-36900Transcriptomic analysis reveals high ITGB1 expression as a predictor for poor prognosis of pancreatic cancerPLOS ONE

Dear Dr. Hoshino,

Thank you for submitting your manuscript to PLOS ONE. After careful consideration, we feel that it has merit but does not fully meet PLOS ONE’s publication criteria as it currently stands. Therefore, we invite you to submit a revised version of the manuscript that addresses the points raised during the review process.<ul> <li>  

1. In your covering letter, you state that ‘there is a lack of transcriptomic data for pancreatic cancer tissues from the East Asian and Japanese population. Hence, in this study, we performed next generation sequencing of pancreatic cancer samples from the Japanese patients, and further validated the sequencing results with immunohistochemical analysis’.

However, the gene that you have chosen to follow up on (ITGB1) has been specifically studied in a Japanese population (which you cite). Taniuchi K, Furihata M, Naganuma S, Sakaguchi M, Saibara T. Overexpression of PODXL/ITGB1 and BCL7B/ITGB1 accurately predicts unfavourable prognosis compared to the TNM staging system in postoperative pancreatic cancer patients. PLOS ONE. 2019;14: e0217920.

I cannot find a clear statement in the methods section as to the name of the hospital from which your samples were obtained. Is there overlap between your patients/samples from those of the Taniuchi PLOS ONE. 2019 study?

Please state exactly where your samples are form in the Methods Section.

2. Please ensure that all data contained or described in the manuscript are made available. PLOS ONE: accelerating the publication of peer-reviewed science.

 </li></ul>

Please submit your revised manuscript by Feb 11 2022 11:59PM. If you will need more time than this to complete your revisions, please reply to this message or contact the journal office at plosone@plos.org. Please include the following items when submitting your revised manuscript:

We look forward to receiving your revised manuscript.

Kind regards,

Eithne Costello

Academic Editor

PLOS ONE

Journal Requirements:

Reviewers' comments:

Reviewer's Responses to Questions

**Comments to the Author**

1. Is the manuscript technically sound, and do the data support the conclusions?

Reviewer #1: Yes

Reviewer #2: Partly

2. Has the statistical analysis been performed appropriately and rigorously? 

Reviewer #1: Yes

Reviewer #2: Yes

3. Have the authors made all data underlying the findings in their manuscript fully available?

Reviewer #1: No

Reviewer #2: Yes

4. Is the manuscript presented in an intelligible fashion and written in standard English?

Reviewer #1: Yes

Reviewer #2: Yes

5. Review Comments to the Author

Reviewer #1: PLOS one

Transcriptomic analysis reveals high ITGB1 expression as a predictor for poor prognosis of pancreatic cancer.

Using transcriptomic analysis Iwatate et al. have demonstrated that ITGB1 is highly expressed in a subset of patients with pancreatic cancer and this expression correlates with a poor prognosis. The authors used immunohistochemical analysis to support this finding.

The authors, state that ITGB1 is ‘an important gene for pancreatic cancer metastasis, progression and prognosis’ and that it ‘may be used as a drug target for pancreatic cancer’. These statements are somewhat supported by the data shown here, and is backed up by previously published work from other groups, but more in vitro and in vivo work would ultimately be needed to test these theories.

This work isn’t novel as it has previously been shown that ITGB1 is associated with poor survival previously (Sun et al. 2018), however, I appreciate that the work done here has been carried out on the under-sequenced Japanese population.

I have outlined some major and minor points below that would need to be addressed before this manuscript is accepted.

Major points

• Tables 1, 2 and 3 are missing from the manuscript and would need to be reviewed before acceptance.

• Concurrent with recent work by Sun et al. 2018 (Prognostic value of increased integrin-beta 1 expression in solid cancers; a meta-analysis). It would be good to show on the data set here the correlation with ITGB1 and OS (this may be shown in the missing figures), and discuss any differences.

• Figure 3. Images are of poor quality, with no scale bar making it difficult to interpret what is happening. The legend says that magnification was 160X – this doesn’t look to be correct. More description both on the figure and in the legend is required here.

• Figure 3. It had been mentioned that the tumour cell IHC scores were between 0-18. This data is not shown. Which sample set was this carried out on? The retrospective 107 patient samples or the 17 samples that were sequenced. Additionally, according to the methods, the maximum histoscore that could be achieved would be >80% (5) x staining higher than the control (3) = 15. However in the next the scores were between 0-18. This needs clarifying.

Minor points

• Ethics statement is missing from the author checklist (although it is present in the materials and methods.

• Sequencing data is not publicly available, with no reason given.

• ‘Transcriptomic analysis of pancreatic cancer samples from the East Asian and Japanese population are lacking’. I appreciate that this is a novel data set that has been sequenced here, however, how does it compare to previous data sets? Are there specific genes/pathways differentially expressed here? Could this set be used for this?

• I’m not sure why the ‘traditional’ method of determining a histoscore wasn’t used here (intensity of 1-3 and % of tissue stained, giving a maximum histoscore of 300, compared to 15).

• Figure 2. Text mentions that ITGB1 was significantly correlated with the prognosis of pancreatic cancer and COL4A1 and COL4A2 were not. Data isn’t shown.

• Figure 4. The r values for the correlation between the RNA-seq and the IHC do not match up between figure and text (p=0.552 and p=0.542).

• Figure 4. It is unclear to what data the p values are relating. Additionally, 9.07 was mentioned on the graph, with no explanation in the figure legend. I’m assuming mean expression?

• Discussion – difference between the authors paper and that of Bailey et.al; the authors claim that the difference may be due to differences in the stromal content of the samples. Samples in the Bailey paper had a high epithelial content ≥40% (not 50% as stated in this manuscript). What is the stromal content of the samples used here?

• Overall the paper would benefit from being proof read

Typos

• RNA sequencing; the authors referred to a p value of 1/10000. The standard nomenclature would be p<0.0001

• IHC scoring of ITGB1 and related definitions; ‘Cases with an IHC expression score higher than the mean RNA expression level of ITGB1… Here it should be italicised due to references RNA expression.

• Patient backgrounds; IPMC is mentioned fir the first time, but hasn’t been defined.

• RNA sequencing; ‘Cytoscape was performed on these DEGs and the hub gene was found to be ITGB1…’ Here it should be italicised. And again in the last sentence of this section.

• Discussion; ‘The study for transcriptome inby Bailey et al.’ ‘inby’ as a typo and et al should be italicised.

Reviewer #2: The work submitted by Iwatate et al. evaluates the role of ITGB1 as a prognostic marker for pancreatic cancer using clinical data as well as transcriptomic analysis by RNAseq and protein expression by IHC. Results are interesting and promising, particularly due to the limited access to clinical samples especially for this type of tumour. Findings are in line with previous work published by others on ITGB1, including other types of cancer, adding value to results obtained using a Japanese cohort. The methodology used is, in general, appropriate and the conclusion is supported by the results presented.

There are, however, a number of points which would need clarification before publication.

-Some extra information on ITGB1 (e.g. its biological function, role/dysregulation in cancer, scheme on pathways regulated by ITGB1 etc) would need to be included in the introduction section to facilitate the understanding of the manuscript. Some general ideas appear in the discussion, but further details need to be added to the introduction, including its clinical relevance.

-Sadly, I could not find Tables 1-3 mentioned in the manuscript in the online system nor in the PDF of the manuscript. Please double check Tables are included in the main text as they are key to follow the results section. Did the authors classify their samples on classical or basal-like types to correlate this with the RNAseq results? Did the authors indicate the stage of disease in the table and its correlation with ITBG1 expression? And the type of postoperative adjuvant chemotherapy?

-All figure legends need to be improved for the readers to be able to understand the figures.

-Results: The authors acknowledge as a limitation of their study the low number of samples analyzed by RNAseq. Different publications have supported the idea that RNAseq can also be performed in fixed tissues (e.g. https://pubmed.ncbi.nlm.nih.gov/31059554/, among others). Because a good correlation between RNAseq data and IHC results could not be accomplished in this study. Did the authors try to perform RNAseq in fixed tissue samples to evaluate if the results were more similar to the obtained IHC data? Please, if possible, add these data in the revised manuscript, and elaborate this in the discussion section, indicating if the observed differences could have been due to comparing RNAseq results obtained from fresh or frozen tissue versus formalin-fixed tissue for IHC. Compare these results with other published data.

-Please include a heatmap showing the RNAseq results for the 12 samples analysed.

-R2 platform analysis (page 11): did the authors also examine ITGAV, COLA1 and COLA2? Please include these details as done for the rest of DEGs.

-Figure 2: Please include Kaplan Meier curves for the rest of genes, not only for ITGB1.

-Quantification of ITGB1 level and intensity in IHC seems to be very dependent on the pathologist evaluating the samples. It is not clear if this could have had an impact on the results obtained by the authors. Did the authors think of performing western blotting to quantify protein expression in their samples and validate if they match the IHC data?

-Fig 3 IHC: the quality of the image showing staining in tumour tissue and adjacent normal pancreas tissue is not great. Please revise it and include further samples in the revised manuscript. For example, samples from different stages of the disease, with high and low ITGB1expression etc. Scale bars are missing in Fig 3. Please include them.

-The Discussion section would benefit from extra information on current and potential/promising prognostic markers for pancreatic cancer used in the clinic (including further details on CA19-9, such as what are considered normal or high levels etc). Also, please expand the information on clinical trials targeting integrins (page 17), including NCT number.

Minor comments:

-Acronyms need to be explained in the text the first time they appear (e.g. ECM, ITGB1, COL etc)

-Abstract: please clarify that the 12 samples analysed by RNAseq were fresh or frozen samples and not fixed tissue obtained from naïve (untreated) patients. Also please indicate that immunostaining was perform only for ITGB1 and not for all the genes identified by RNAseq.

-Page 7, line 6: please indicate what clinicopathological factors authors are referring to.

-Please indicate the concentration of the ITGB1 antibody used for IHC stainings.

-Please explain what T-factors or T2 are (page 11).

-Page 17, line 15: please clarify what the authors mean by “by case stratification”. Also, this affirmation “Although the influence of ITGB1 in pancreatic cancer has not been reported, it is expected that ITGB1 will contribute to markers and treatment in pancreatic cancer” is not clear as the authors previously highlight that the role of ITGB1 in pancreatic cancer has been previously described by others.

-Please revise the English grammar before re-submitting the manuscript. Thank you.

6. PLOS authors have the option to publish the peer review history of their article (what does this mean?). If published, this will include your full peer review and any attached files.

Reviewer #1: No

Reviewer #2: No

---

## [Author Response · Author response to Decision Letter 0]

11 Feb 2022

Dear Dr. Hoshino,

Thank you for submitting your manuscript to PLOS ONE. After careful consideration, we feel that it has merit but does not fully meet PLOS ONE’s publication criteria as it currently stands. Therefore, we invite you to submit a revised version of the manuscript that addresses the points raised during the review process.

• 

1. In your covering letter, you state that ‘there is a lack of transcriptomic data for pancreatic cancer tissues from the East Asian and Japanese population. Hence, in this study, we performed next generation sequencing of pancreatic cancer samples from the Japanese patients, and further validated the sequencing results with immunohistochemical analysis.

However, the gene that you have chosen to follow up on (ITGB1) has been specifically studied in a Japanese population (which you cite). Taniuchi K, Furihata M, Naganuma S, Sakaguchi M, Saibara T. Overexpression of PODXL/ITGB1 and BCL7B/ITGB1 accurately predicts unfavourable prognosis compared to the TNM staging system in postoperative pancreatic cancer patients. PLOS ONE. 2019;14: e0217920.

→Comments from our author 

Thank you very　much　for your understanding.　

As you said, I feel that you are right.　

We knew that large-scale comprehensive research using next-generation sequencers was being conducted in the world (in U.S.A., Australia and so on) for pancreatic cancer. However, since there are few data using the next-generation sequencer for pancreatic cancer, especially in Japan, including East Asia, we thought that the study in the population in Japan would be useful this time. Furthermore, we decided to set a cutoff by clarifying the correlation between the results of the next-generation sequencer and the expression of immunostaining. As a result, we believe that the results of immunostaining were obtained as an evaluation with an objective quantitative scale rather than the evaluation of qualitative results such as the presence or absence of staining. The results were similar to those of Taniuchi et al., But by quantifying them using a next-generation sequencer or immunostaining, the usefulness of ITGB1 could be supplemented as a more objective result than the results of Taniuchi et al. I am confident that I was able to do it.

・I cannot find a clear statement in the methods section as to the name of the hospital from which your samples were obtained. Is there overlap between your patients/samples from those of the Taniuchi PLOS ONE. 2019 study?

Please state exactly where your samples are form in the Methods Section.

→Comments from our author

Thank you very much for your suggestion.

I mentioned in the Materials and Methods section that all samples were taken at the Chiba Cancer Center.

Reviewer's Responses to Questions

Comments to the Author

Reviewer #1: PLOS one

Transcriptomic analysis reveals high ITGB1 expression as a predictor for poor prognosis of pancreatic cancer.

Using transcriptomic analysis Iwatate et al. have demonstrated that ITGB1 is highly expressed in a subset of patients with pancreatic cancer and this expression correlates with a poor prognosis. The authors used immunohistochemical analysis to support this finding.

The authors, state that ITGB1 is ‘an important gene for pancreatic cancer metastasis, progression and prognosis’ and that it ‘may be used as a drug target for pancreatic cancer’. These statements are somewhat supported by the data shown here, and is backed up by previously published work from other groups, but more in vitro and in vivo work would ultimately be needed to test these theories.

This work isn’t novel as it has previously been shown that ITGB1 is associated with poor survival previously (Sun et al. 2018), however, I appreciate that the work done here has been carried out on the under-sequenced Japanese population.

I have outlined some major and minor points below that would need to be addressed before this manuscript is accepted.

Major points

• Tables 1, 2 and 3 are missing from the manuscript and would need to be reviewed before acceptance.

→Comments from our author

Thank you for your advice. I must sincerely apologize.

We have added Tables 1, 2 and 3.

• Concurrent with recent work by Sun et al. 2018 (Prognostic value of increased integrin-beta 1 expression in solid cancers; a meta-analysis). It would be good to show on the data set here the correlation with ITGB1 and OS (this may be shown in the missing figures), and discuss any differences.

→Comments from our author

Thank you for your suggestion.

We quoted the "Quanwu Sun, et al Prognostic value of increased integrin-beta 1 expression in solid cancers: a meta-analysis. Onco Targets Ther. 2018; 11: 1787–1799.” meta-analysis and added it to the Discussion as follows. We have also added a figure (Fig. 5) for the results of our OS and DFS.

A meta-analysis was performed by summarizing these studies [27].The meta-analysis that summarized reports of the association between ITGB1 expression and prognosis. Among the accumulated reports, two reports of immunohistochemical staining for pancreatic cancer were found [25,26]. Of these, the study by Sawai et al. used 78 pancreatic cancer patient specimens and investigated the association between ITGB1 expression and prognosis by immunohistological staining [25]. Their results, unlike ours, did not show a significant correlation between ITGB1 expression and prognosis. However, in their report, about 20 postoperative cases of stage IV simultaneous liver metastasis were included, and the background of the patients was significantly different from that of ours, which targeted radical resection cases. Also, the method of evaluating immunohistological staining was different between us and them. Yang et al. Targeted only R0, R1 resectable pancreatic cancer, and our study was consistent with the target cases [26]. In addition, as a result of investigating the relationship between ITGB1 expression and prognosis in 63 cases, the prognosis was poor in the high expression group as in our result [26]. In our study, the number of target cases was about twice as many, and the results conformed to their results.

• Figure 3. Images are of poor quality, with no scale bar making it difficult to interpret what is happening. The legend says that magnification was 160X – this doesn’t look to be correct. More description both on the figure and in the legend is required here.

→Comments from our author

Thank you for your advice.

Also, I am very sorry. The micrographs of high and low expression of ITGB1 subjected to immunostaining were 160 X, and the contrast photographs with the adjacent normal pancreatic tissue were 40 X. I have attached a scale for the sake of clarity.

Also, the deterioration of image quality is considered to be a problem in the journal review process. The image quality of the original figure can be sufficiently confirmed to the cellular level. If necessary, you need to contact the publisher to confirm the image quality of the figure.

• Figure 3. It had been mentioned that the tumour cell IHC scores were between 0-18. This data is not shown. Which sample set was this carried out on? The retrospective 107 patient samples or the 17 samples that were sequenced. Additionally, according to the methods, the maximum histoscore that could be achieved would be >80% (5) x staining higher than the control (3) = 15. However in the next the scores were between 0-18. This needs clarifying.

→Comments from our author

Thank you for your advice. 

All cases of ITGB1 immunostaining have been successfully performed. We are very sorry. 100% = 6. I did not mention it, so I added it　in manuscript.

Minor points

• Ethics statement is missing from the author checklist (although it is present in the materials and methods.

→Comments from our author

Thank you very much. We have added an Ethics Statement to the checklist.

• Sequencing data is not publicly available, with no reason given.

→Comments from our author

Thank you very much. We are uploading RNA-seq results to Gene Expression Omnibus. The 　accession number is GSE196009. 

• ‘Transcriptomic analysis of pancreatic cancer samples from the East Asian and Japanese population are lacking’. I appreciate that this is a novel data set that has been sequenced here, however, how does it compare to previous data sets? Are there specific genes/pathways differentially expressed here? Could this set be used for this?

→Comments from our author

RAW data cannot be obtained from TCGA, etc. at this facility due to ethical review. Processing data is public data and can be obtained. Since it is not raw data, it is not possible to actually calculate the Differential Expression Genes (DEGs) between this study and the TCGA study. In addition, TCGA has only 4 data on adjacent pancreatic tissue, which makes it difficult to calculate the same amount of DEGs as we do. Although it is microarray data, there is a meta-analysis of DEGs identification and pathway analysis by comparison between normal pancreas and pancreatic cancer as in our study. In that study, the integrin family and collagen were identified as in the results of this study, and they are also accumulated in ECM-Receptor Interaction, PI3K-Akt Signaling Pathway, Focal Adhesion, Pathways in Cancer, Pancreatic Secretion, Metabolic Pathways, etc. This is partly consistent with the research in.

Sevcan Atay Integrated transcriptome meta-analysis of pancreatic ductal adenocarcinoma and matched adjacent pancreatic tissues. PeerJ 8: e10141 DOI 10.7717 / peerj.10141

Vandana Sandhu, PhD et al. Meta-Analysis of 1,200 Transcriptomic Profiles Identifies a Prognostic Model for Pancreatic Ductal Adenocarcinoma, Clin Cancer Inform. 2019: DOI https://doi.org/10.1200/CCI.18.00102

• I’m not sure why the ‘traditional’ method of determining a histoscore wasn’t used here (intensity of 1-3 and % of tissue stained, giving a maximum histoscore of 300, compared to 15).

→Comments from our author

Thank you very much. We think it is reasonable that you have pointed out. We also considered using the classic H score (% * intensity) you pointed out, but this time we have referred to　other scoring : e.g. Allred scores and immune responses score (IRS). We used scoring that divides every 20% with reference to these scoring.　Nickolay Fedchenko Janin Reifenrath　： Different approaches for interpretation and reporting of immunohistochemistry analysis results in the bone tissue - a review　 Diagn Pathol.　2014 Nov 29;9:221. doi: 10.1186/s13000-014-0221-9.

• Figure 2. Text mentions that ITGB1 was significantly correlated with the prognosis of pancreatic cancer and COL4A1 and COL4A2 were not. Data isn’t shown.

→Comments from our author

Thank you very much. We added COL4A2 and COL4A2 Kaplan-Maier curves to Fig.2.

• Figure 4. The r values for the correlation between the RNA-seq and the IHC do not match up between figure and text (p=0.552 and p=0.542).

→Comments from our author

Thank you for your advice.

Since r = 0.552, We changed the figure accordingly.

• Figure 4. It is unclear to what data the p values are relating. Additionally, 9.07 was mentioned on the graph, with no explanation in the figure legend. I’m assuming mean expression?

→Comments from our author

Thank you very much. As you said, 9.07 is the average expression level. However, due to data variability, we have adopted the median nonparametric this time. We corrected it and added it to Figure legends. And in order to evaluate the significance of the correlation between the RNA expression level of ITGB1 using RNA-seq and the IHC score of ITGB1 using immunohistochemistry, we calculated the Spearman’s rank correlation coefficient (r, ρ). We have described these in the sections "Correlation between IHC score and RNA-sequencing" and "Definitions of variables for clinicopathological factors and statistical analysis".

• Discussion – difference between the authors paper and that of Bailey et.al; the authors claim that the difference may be due to differences in the stromal content of the samples. Samples in the Bailey paper had a high epithelial content ≥40% (not 50% as stated in this manuscript). What is the stromal content of the samples used here?

→Comments from our author

Thank you very much. Pancreatic cancer has long been said to be a tumor with many stroma, which has made it difficult to determine expression data and genomic data, but Bailey et al. presented more reliable data by extracting the genome from the many tissues with few stroma and many tumor cells. 

We are sorry for the lack of words. As you said, we think so too.

What Bailey et al. performed is a deep sequence of exome, which only clusters RNA transcriptional networks and RNA expression. The reason for the difference in pathway between the results of exome by Bailey et al. and our mRNA expression is that since we are comparing cancer tissue with adjacent normal pancreas and cancer tissue has more stromal tissue than normal pancreas, we believe that the results of the stromal tissue may be over-reflected.

The metadata of the expression analysis including the microarray shows the same pathway analysis result as ours.

I would appreciate it if you could refer to it.

We also collected samples from the part of the specimen where the high content of tumor cells was confirmed to be 40% or more by HE staining, and sequenced them.

Sevcan Atay Integrated transcriptome meta-analysis of pancreatic ductal adenocarcinoma and matched adjacent pancreatic tissues. PeerJ 8: e10141 DOI 10.7717 / peerj.10141

Vandana Sandhu, PhD et al. Meta-Analysis of 1,200 Transcriptomic Profiles Identifies a Prognostic Model for Pancreatic Ductal Adenocarcinoma, Clin Cancer Inform. 2019: DOI https://doi.org/10.1200/CCI.18.00102

Based on the above, we have rewritten the discussion.

• Overall the paper would benefit from being proof read

→Comments from our author

Thank you. We requested English proofreading again.

Typos

• RNA sequencing; the authors referred to a p value of 1/10000. The standard nomenclature would be p<0.0001

→Comments from our author

Thank you very much. We have corrected it as above.

• IHC scoring of ITGB1 and related definitions; ‘Cases with an IHC expression score higher than the mean RNA expression level of ITGB1… Here it should be italicised due to references RNA expression.

→Comments from our author

Thank you very much. We have corrected the all "ITGB1" in the section of “IHC scoring of ITGB1 and related definitions” to italics.

• Patient backgrounds; IPMC is mentioned fir the first time, but hasn’t been defined.

→Comments from our author

Thank you very much. Regarding the first IPMC we mentioned, we have made the following corrections.

IPMC　→　intraductal papillary mucinous carcinoma (IPMC)

• RNA sequencing; ‘Cytoscape was performed on these DEGs and the hub gene was found to be ITGB1…’ Here it should be italicised. And again in the last sentence of this section.

→Comments from our author

Thank you very much. We have corrected the pointed out "ITGB1" to italics.

• Discussion; ‘The study for transcriptome inby Bailey et al.’ ‘inby’ as a typo and et al should be italicised.

→Comments from our author

Thank you very much. We have corrected as above.

Reviewer #2: The work submitted by Iwatate et al. evaluates the role of ITGB1 as a prognostic marker for pancreatic cancer using clinical data as well as transcriptomic analysis by RNAseq and protein expression by IHC. Results are interesting and promising, particularly due to the limited access to clinical samples especially for this type of tumour. Findings are in line with previous work published by others on ITGB1, including other types of cancer, adding value to results obtained using a Japanese cohort. The methodology used is, in general, appropriate and the conclusion is supported by the results presented.

There are, however, a number of points which would need clarification before publication.

-Some extra information on ITGB1 (e.g. its biological function, role/dysregulation in cancer, scheme on pathways regulated by ITGB1 etc) would need to be included in the introduction section to facilitate the understanding of the manuscript. Some general ideas appear in the discussion, but further details need to be added to the introduction, including its clinical relevance.

→Comments from our author

Thank you for your suggestion. We have added the following to the introduction.

ITGB1 is a constituent of β subunits in integrin molecules [9]. Integrin is mainly present in the plasma membrane and plays role of cell-cell adhesion, cell-extracellular matrix adhesion, and signal transduction [9, 10]. These lacks of adhesion cause the withdrawal of cell survival signals, resulting in exfoliation-induced apoptosis called "anoikis"[11]. Cancer cells are resistant to anoikis through certain types of integrins and are recognized as one of the key mechanisms for successful infiltration, migration and metastasis [11]. It has been reported that high expression of ITGB1 significantly correlates with deterioration of prognosis in colorectal cancer, breast cancer, lung cancer, etc., but it is controversial in pancreatic cancer[12-17].

-Sadly, I could not find Tables 1-3 mentioned in the manuscript in the online system nor in the PDF of the manuscript. Please double check Tables are included in the main text as they are key to follow the results section. Did the authors classify their samples on classical or basal-like types to correlate this with the RNAseq results? Did the authors indicate the stage of disease in the table and its correlation with ITBG1 expression? And the type of postoperative adjuvant chemotherapy?

→Comments from our author

Thank you for your support. I am very sorry. Table 1-3 was not attached on the previous manuscript, so we will attach it with this revise. We did not classify it as a classic type or a base-like type. Instead, we examined the correlation between the RNA expression level quantified by count per million (CPM) and the expression level by immunostaining in ITGB1 using Spearman's rank correlation coefficient. Postoperative chemotherapy is only with or without S-1. We also show this in Table 1.　

-All figure legends need to be improved for the readers to be able to understand the figures.

→Comments from our author

Thank you for your suggestion.　All the legends have been rewritten in detail. The legend of the figure has been greatly revised. We would appreciate it if you could accept it.

-Results: The authors acknowledge as a limitation of their study the low number of samples analyzed by RNAseq. Different publications have supported the idea that RNAseq can also be performed in fixed tissues (e.g. https://pubmed.ncbi.nlm.nih.gov/31059554/, among others). Because a good correlation between RNAseq data and IHC results could not be accomplished in this study. Did the authors try to perform RNAseq in fixed tissue samples to evaluate if the results were more similar to the obtained IHC data? Please, if possible, add these data in the revised manuscript, and elaborate this in the discussion section, indicating if the observed differences could have been due to comparing RNAseq results obtained from fresh or frozen tissue versus formalin-fixed tissue for IHC. Compare these results with other published data.

→Comments from our author

Thank you for your advice.

We think that you are right. We have previously attempted RNA extraction from FFPE. However, mRNA prepared from FFPE older than 1 year could hardly meet the quality check, and mRNA suitable for the conditions could be extracted from frozen specimens older than 1 year. This is probably due to pancreatic proteolytic enzymes and specimen deterioration depends on storage conditions.　Currently, it has passed since the time of the experiment, and all FFPEs have passed more than one year.

No additional verification was possible. As you said, We think it is undeniable that the difference between FFPE and frozen specimens may lead to the discrepancy of mRNA expression. These pursuits will be our future task.

-Please include a heatmap showing the RNAseq results for the 12 samples analysed.

→Comments from our author

Thank you very much. We detected 314 DEGs in 17 samples of 12 patients, created these heatmaps, and added them to the supplement (S1 Fig).

-R2 platform analysis (page 11): did the authors also examine ITGAV, COL1A1 and COL1A2? Please include these details as done for the rest of DEGs.

→Comments from our author

Thank you very much.

The rest of DEGs genes also analyzed this　time were COL1A1, COL1A2, and ITGAV and so on, so we verified these on the R2 platform. Only ITGAV was significantly correlated with prognosis, (COL1A1, COL1A2, and ITGAV,　P　=　0.098,0.174,0.033,respectively)　but this time it was not included in the analysis because it was not a high-ranking hub gene.

-Figure 2: Please include Kaplan Meier curves for the rest of genes, not only for ITGB1.

→Comments from our author

Thank you very much.

Second to the top hub genes analyzed this time were ITGB1, COL4A1, COL4A2, and ITGA5, so we verified these on the R2 platform and added them to Fig2.

-Quantification of ITGB1 level and intensity in IHC seems to be very dependent on the pathologist evaluating the samples. It is not clear if this could have had an impact on the results obtained by the authors. Did the authors think of performing western blotting to quantify protein expression in their samples and validate if they match the IHC data?

→Comments from our author

Thank you very much. As you pointed out, We feel that you are right.

When it comes to protein quantification, we think it makes more sense to do a Western plot. However, we could not extract mRNA with satisfactory quality except for cryopreserved specimens and raw specimens. And currently, half of the frozen and raw specimens have been used up for RNA extraction. We also tried to do it with FFPE, but we couldn't do it because we didn't have any experienced staff to do Western plot from FFPE in the pancreas. We would like to keep these issues as future issues.

-Fig 3 IHC: the quality of the image showing staining in tumour tissue and adjacent normal pancreas tissue is not great. Please revise it and include further samples in the revised manuscript. For example, samples from different stages of the disease, with high and low ITGB1expression etc. Scale bars are missing in Fig 3. Please include them.

→Comments from our author

Thank you for your advice.

The image quality has been improved, including photographs of tumor tissue and adjacent normal pancreatic tissue, and high-quality histological photographs with high and low expression of ITGB1 have been added. We have also added a scale to the tissue photo.

-The Discussion section would benefit from extra information on current and potential/promising prognostic markers for pancreatic cancer used in the clinic (including further details on CA19-9, such as what are considered normal or high levels etc). Also, please expand the information on clinical trials targeting integrins (page 17), including NCT number.

→Comments from our author

Thank you very much.

We feel that you are right. There is a certain opinion about the tumor marker (CA19-9 CEA) that is generally known for pancreatic cancer and its prognostic significance, and we compared it with our data with references and entered it in "Discussion" as follows.

The known preoperative tumor markers CA19-9 and CEA, which are potential prognostic factors, have cutoffs of 37 U / ml and 3 U / ml or 5 U / ml, respectively [26]. It was said that high preoperative marker levels can be a prognostic factor, but they cannot be therapeutic targets. In this study, we divided the median into two groups, but at our facility, the inspection cutoffs for CA19-9 and CEA were 37U / ml and 5.0U / ml, respectively. When examined, the high value of CA19-9 was significantly correlated with DFS (OS; CA19-9＞37.0, CEA＞3,0, CEA＞5,0 P = 　0.051, 0.079, 0.233, respectively, DFS; CA19-9＞37.0, CEA＞3,0, CEA＞5,0 P = 0.001, 0.286, 0.356, respectively), but in multivariate analysis, CA19-9 was not an independent factor (P = 0.141). 

Marius Distler　et al. Preoperative CEA and CA 19-9 are prognostic markers for survival after curative resection for ductal adenocarcinoma of the pancreas – A retrospective tumor marker prognostic study　 International Journal of Surgery, 2013-12-01, Volume 11, Issue 10, Pages 1067-1072

Regarding the results and status of clinical trials of ITGB1, we have added a description including the NCT number to "Discussion" and further references as shown below.

In clinical trials, there are currently no reports showing that a single integrin inhibitor is effective, but they are expected to be effective in combination with multiple agents such as immune checkpoint inhibitors (NCT00195278, NCT04508179) [39, 40]

R J Slack et al. Emerging therapeutic opportunities for integrin inhibitors　 Nat Rev Drug Discov. 2022 Jan;21(1):60-78. doi: 10.1038/s41573-021-00284-4.

Minor comments:

-Acronyms need to be explained in the text the first time they appear (e.g. ECM, ITGB1, COL etc)

→Comments from our author

Thank you for your advice. We have explained acronyms such as ECM, PI3K-Akt, ITGB1, COL, ITGA5, ITGAV, etc. without using abbreviations.

-Abstract: please clarify that the 12 samples analysed by RNAseq were fresh or frozen samples and not fixed tissue obtained from naïve (untreated) patients. Also please indicate that immunostaining was perform only for ITGB1 and not for all the genes identified by RNAseq.

→Comments from our author

Thank you very much. We emphasized in the "Abstract" section that we did not use fixative tissue and that immunostaining was performed only on ITGB1.

-Page 7, line 6: please indicate what clinicopathological factors authors are referring to.

→Comments from our author

Thank you very much for your point out.

It was written in a bad way, so I rewrote it drastically.

-Please indicate the concentration of the ITGB1 antibody used for IHC stainings.

→Comments from our author 

Thank you very much.

According to the instruction manual, it is diluted 100 times. I have described it in "Materials and Methods".

-Please explain what T-factors or T2 are (page 11).

→Comments from our author

Thank you very much.

We explained about the T factor.

-Page 17, line 15: please clarify what the authors mean by “by case stratification”. Also, this affirmation “Although the influence of ITGB1 in pancreatic cancer has not been reported, it is expected that ITGB1 will contribute to markers and treatment in pancreatic cancer” is not clear as the authors previously highlight that the role of ITGB1 in pancreatic cancer has been previously described by others.

→Comments from our author

Thank you for your advice.

“By case stratification” was an ambiguous expression, so we removed it.

Similarly, the expression “Although the influence of ITGB1 in pancreatic cancer has not been reported,” was deleted because it was considered inappropriate.

-Please revise the English grammar before re-submitting the manuscript. Thank you.

→Comments from our author

Thank you. We requested English proofreading again.

---

## [Decision Letter · Decision Letter 1]

4 May 2022

Transcriptomic analysis reveals high ITGB1 expression as a predictor for poor prognosis of pancreatic cancer

PONE-D-21-36900R1

Dear Dr. Hoshino,

I wish to apologise for the delay in providing you with a decision, and I thank you for your patience.

We’re pleased to inform you that your manuscript has been judged scientifically suitable for publication and will be formally accepted for publication once it meets all outstanding technical requirements.

Kind regards,

Eithne Costello

Academic Editor

PLOS ONE

Additional Editor Comments (optional):

Thank you for the many modifications you have made to your manuscript.

May I please ask that you ensure that *supplementary (S1 Fig) *is included in the resubmission, because I cannot find it in the current version.

Secondly, reviewer one points out a minor typographical error, that I would ask you to correct, i.e. please change *integinand* to *integrin and*. 

I am happy with the quality of figure 3, thank you for your modifications to that.

Reviewers' comments:

Reviewer's Responses to Questions

**Comments to the Author**

1. If the authors have adequately addressed your comments raised in a previous round of review and you feel that this manuscript is now acceptable for publication, you may indicate that here to bypass the “Comments to the Author” section, enter your conflict of interest statement in the “Confidential to Editor” section, and submit your "Accept" recommendation.

Reviewer #1: (No Response)

2. Is the manuscript technically sound, and do the data support the conclusions?

Reviewer #1: Yes

3. Has the statistical analysis been performed appropriately and rigorously? 

Reviewer #1: Yes

4. Have the authors made all data underlying the findings in their manuscript fully available?

Reviewer #1: Yes

5. Is the manuscript presented in an intelligible fashion and written in standard English?

Reviewer #1: No

6. Review Comments to the Author

Reviewer #1: The authors have addressed all my concerns, with the exception of the image quality of figure 3.

The authors propose that the image quality is ‘considered to be a problem in the journal review process’ that I should contact the publisher to confirm image quality.

I’ve not received a reply from the publisher regarding this figure, so cannot comment on the material within this image or conclusions derived from it.

Typo;

Page 24 in the discussion;

While it was reported that it functioned as a cell by ‘’construction of scaffolds’’ with integinand

Change in integrin and…

If the issue with image quality is resolved, I’d be happy to take another quick look at this section and recommend it for publication.

7. PLOS authors have the option to publish the peer review history of their article (what does this mean?). If published, this will include your full peer review and any attached files.

Reviewer #1: No

---

## [Editor Report · Acceptance letter]

23 May 2022

PONE-D-21-36900R1 

Transcriptomic analysis reveals high *ITGB1* expression as a predictor for poor prognosis of pancreatic cancer 

Dear Dr. Hoshino:

I'm pleased to inform you that your manuscript has been deemed suitable for publication in PLOS ONE. Congratulations! Your manuscript is now with our production department. 

Kind regards, 

on behalf of

Dr. Eithne Costello 

Academic Editor

PLOS ONE